# WinDeskGround: A Benchmark for Robust GUI Grounding in Complex Multi-Window Desktop Environments

Haoren Zhao [* 1]    Tianyi Chen [* 2]    Zhen Wang [1]

## Abstract

Multimodal Large Language Models (MLLMs) have revolutionized GUI automation, yet their efficacy is largely established on idealized, single-layer interfaces. This paper identifies a critical reliability gap: state-of-the-art agents face distinct robustness challenges in real-world desktop environments characterized by multi-window stacking, occlusion, and visual clutter. To address this, we introduce WinDeskGround, a novel benchmark and synthesis framework tailored for evaluating GUI grounding robustness. Unlike static datasets, our framework parametrically generates complex desktop scenarios by controlling window occlusion, layout density, and semantic similarity, thereby simulating the distribution shifts of authentic workflows. We construct a diverse meta-dataset of 1,356 high-fidelity instruction-target pairs and conduct comprehensive evaluations of five leading MLLMs. Our results demonstrate that while top-tier agents excel in simplified settings, their accuracy declines under partial occlusion. WinDeskGround provides a valuable benchmark to facilitate the assessment and advancement of GUI agent robustness in realistic environments. The code is available at https://github.com/ZZZhr-1/WinDeskGround.

## 1. Introduction

With the rapid advancement of Multimodal Large Language Models (MLLMs) (Hurst et al., 2024; Bai et al., 2025), GUI automation has undergone a paradigm shift (Zhang et al., 2024; Sager et al., 2025). Traditional automation approaches primarily rely on DOM trees, system accessibility APIs (Deng et al., 2023; Liu et al., 2018), or more recent

tool-selection-based frameworks (Chen et al., 2025). However, in cross-platform and complex rendering scenarios, such structured data often faces bottlenecks due to incompleteness or incompatibility (Xu et al., 2024). In contrast, MLLM-based agents can perceive screen information visually, akin to humans, without being constrained by the underlying code structure (Cheng et al., 2024; Chen et al., 2024). This capability, known as GUI Grounding, leverages powerful visual reasoning to map natural language instructions directly to screen pixels, enabling agents to generalize and make decisions in unseen interfaces. Consequently, it has become a core direction in GUI automation research.

The execution of a task by a GUI-based Computer Use Agent (CUA) can be divided into two phases: planning and grounding (Chen et al., 2026; Zhang et al., 2025; Zhao et al., 2025). Planning involves analyzing the task description and current state to determine future actions, whereas grounding refers to the simulation of input device operations to execute these actions (Shlomov et al., 2024). For GUI agents, GUI Grounding is the fundamental capability for perceiving the environment and performing operations. Although this pure vision paradigm has achieved significant success in single-task scenarios on Mobile (Wang et al., 2024) and Web platforms (Pan et al., 2024), it faces distinct challenges in Desktop environments.

Real-world user desktop environments are characterized by high complexity and clutter. Unlike mobile interfaces that typically display a single application in full screen, desktop workflows often involve multi-task parallel processing (Li et al., 2025). This results in multi-window stacking, dense layouts, and interference from visually similar elements, as shown in Figure 1.

Such complex scenes constitute a major bottleneck for GUI semantic positioning models in practical applications. However, existing public datasets (Hui et al., 2025; Wu et al., 2024) primarily consist of screenshots of single application windows or cleaned, idealized interfaces, which fail to reflect model performance under realistic conditions of multi-window stacking, occlusion, and high-density layouts. This distribution shift between training data and real-world application scenarios leads to a substantial decline in the robustness and accuracy of existing models when facing

---

[*]Equal contribution  [1]School of Cyberspace, Hangzhou Dianzi University, Hangzhou, China  [2]Meta AI. Correspondence to: Zhen Wang <wangzhen@hdu.edu.cn>.

*Proceedings of the 43rd International Conference on Machine Learning*, Seoul, South Korea. PMLR 306, 2026. Copyright 2026 by the author(s).

complex desktop environments.

To address this research gap, this paper focuses on evaluating the robustness of GUI Grounding in desktop environments. Addressing the lack of complex layout characteristics in existing data, we propose a multi-window desktop synthesis method. Rather than directly collecting complex desktop data, which is difficult to annotate, we rely on high-fidelity single-window metadata. By parametrically controlling window layout, density, occlusion ratios, and semantic similarity, we automatically generate synthetic test samples with varying levels of difficulty.

This approach not only solves the challenge of acquiring data for complex scenarios but also enables fine-grained evaluation of model adaptability under specific interference conditions (e.g., varying degrees of occlusion). This design also clarifies our novelty relative to recent benchmarks such as ScreenSpot-Pro (Li et al., 2025): although they include a portion of multi-window samples, they treat such complexity as part of the natural data distribution rather than as an explicitly controllable variable. By contrast, we cast multi-window complexity as a parameterized and decomposable evaluation space, enabling systematic factor-level analysis of GUI grounding robustness with respect to layout density, occlusion, and semantic similarity. Furthermore, to support this method, we constructed and open-sourced a meta-dataset covering 9 major application domains (including productivity tools, browsers, development tools, etc.), comprising 585 high-resolution real window screenshots and 1,356 instructions.

The main contributions are summarized as follows:

- **Simulated Multi-window Desktop Synthesis Framework.** We propose a general-purpose framework for simulating multi-window desktop synthesis. This framework simulates the complexity of real-world user desktops and supports the parametric generation of scenarios involving window occlusion, high-density layouts, and visual similarity interference, providing a new benchmark for evaluating the anti-interference capabilities of GUI Agents.

- **WinDeskGround Benchmark.** Based on this synthesis method, we constructed a high-quality window screenshot meta-dataset covering diverse application domains, ranging from office software to development tools. Leveraging this dataset and the synthesis framework, we conducted a comprehensive evaluation and analysis of the GUI Grounding performance of existing MLLMs in simulated desktop scenarios, revealing the limitations of models in complex desktop environments and suggesting directions for improvement.

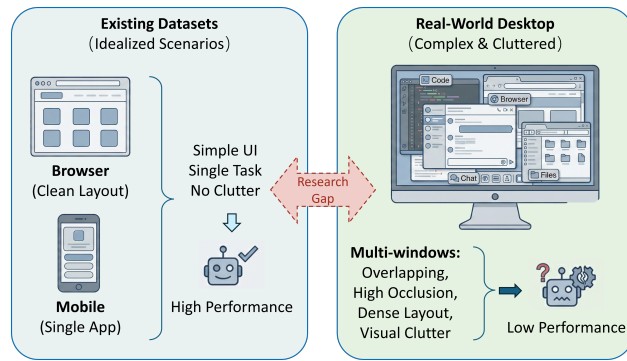

*Figure 1.* **The distribution shift and research gap between existing datasets and real-world desktops.** While existing datasets focus on idealized scenarios (Mobile/Web), real-world desktops exhibit complexity through multi-window stacking and visual clutter, leading to low robustness in out-of-domain settings.

## 2. Related Work

### 2.1. GUI Datasets and Environments

High-quality datasets serve as the foundation for training and evaluating GUI agents. Early research predominantly focused on mobile applications; for instance, Rico (Deka et al., 2017) and AMP (Zhang et al., 2021) provided extensive collections of Android and iOS screenshots paired with metadata, significantly advancing mobile UI understanding. Subsequent studies have expanded into Web and desktop operating systems, with datasets like Mind2Web (Deng et al., 2023) and Visual-WebArena (Koh et al., 2024) offering simulated environments for web-based agents.

However, a significant gap remains regarding the desktop environment. While recent benchmarks such as WindowsArena (Bonatti et al., 2024) and OSWorld (Xie et al., 2024) attempt to replicate real operating system environments, they primarily focus on high-level task planning success rates rather than the fine-grained robustness of UI grounding. Crucially, existing public datasets, such as Screenspot (Cheng et al., 2024; Wu et al., 2024) and WinSpot (Hui et al., 2025), mostly consist of single-application windows or sanitized interfaces. They fail to capture the multi-window stacking, high-density layouts, and visual occlusion that are ubiquitous in real-world user workflows. This distribution shift between training data and realistic scenarios limits the generalization of agents in complex desktop environments. Even ScreenSpot-Pro (Li et al., 2025), which introduces high-resolution desktop screenshots and includes a subset of multi-window scenarios, follows naturally collected distributions and does not provide explicit control over factors such as occlusion ratio, layout density, or semantic similarity; consequently, it is less suitable for controlled analysis of failure modes. In this work, we address this deficiency by introducing a parameterized multi-window synthesis method and a meta-dataset

explicitly designed to evaluate performance under complex desktop conditions.

## 2.2. MLLM-based GUI Agents

The integration of Large Language Models (LLMs) and Multimodal LLMs (MLLMs) has introduced a new paradigm for GUI automation (Brown et al., 2020; Yang et al., 2023b). Traditional approaches typically rely heavily on structured data, such as HTML DOM trees or Accessibility APIs, to parse interface structures (Deng et al., 2023). However, this dependency on structured metadata proves fragile in cross-platform settings or when dealing with legacy software where such APIs are inaccessible or inconsistent.

To circumvent the reliance on underlying system interfaces, the research community has pivoted towards vision-only solutions. Techniques like Set-of-Mark (SoM) (Yang et al., 2023a) overlay visual markers on screenshots to assist models like GPT-4V in grounding; however, this approach still necessitates auxiliary detection models or metadata to generate the markers initially. In contrast, recent end-to-end approaches, such as SeeClick (Cheng et al., 2024) and Fuyu (Bavishi et al., 2023), attempt to directly interpret raw screen pixels. Despite their promising performance on standard benchmarks, these vision-centric models are predominantly evaluated on clean, interference-free interfaces. There remains a lack of systematic research regarding the robustness of MLLMs when effectively deployed in high-clutter, chaotic desktop environments. Our work aims to bridge this gap by simulating complex visual interference scenarios to rigorously evaluate and improve the adaptability of vision-only GUI agents in the wild.

## 3. Methodology

### 3.1. Preliminary and Task Formulation

To evaluate MLLMs in desktop environments, we formalize the GUI grounding task. Let $S$ represent an interface screenshot, and $E = \{e_i = (x_i, y_i) \mid i = 1, \ldots, N\}$ denote the set of actionable GUI elements. Here, $x_i$ represents the textual description (or natural language instruction) of the element, and $y_i$ denotes its corresponding spatial location (typically formatted as a bounding box or a center point).

Our study focuses exclusively on evaluating the GUI Grounding capability. This task simulates the process of an agent executing user instructions: given the screenshot $S$ and a specific textual instruction $x$, the model is required to predict the spatial coordinates $\hat{y}$ of the target element. Formally, this involves inferring the output based on the conditional probability $P(y \mid S, x)$.

In this evaluation, we focus on models that adopt the natural language coordinate paradigm (e.g., Uground (Gou et al.,

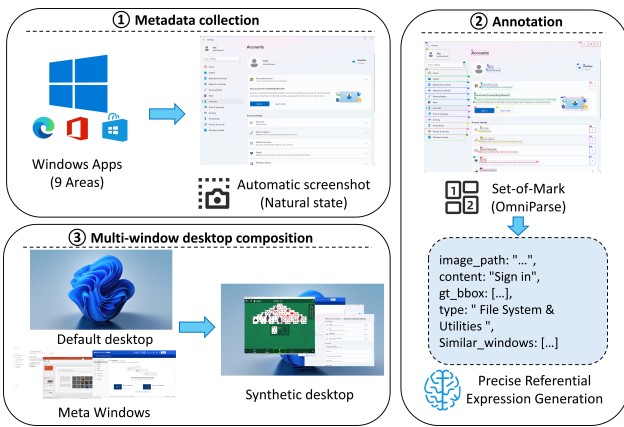

*Figure 2.* The pipeline of data construction for the metadata.

2025), SeeClick (Cheng et al., 2024)). Unlike earlier approaches that treat coordinates as classification bins (Wang et al., 2023), these state-of-the-art models normalize spatial coordinates to a specific range (e.g., $[0, 1]$ or $[0, 1000]$) and process them as plain text sequences. During the inference phase, the model autoregressively generates the target coordinate sequence based on the provided visual and textual context. For a given input (image $S$ and instruction $x$), the model's predicted output $\hat{y}$ is obtained by maximizing the generation probability:

$$\hat{y} = \arg\max_y \prod_{j=1}^{L} P(y_j \mid S, x, y_{<j}) \tag{1}$$

where $y_{<j}$ denotes the coordinate tokens generated in previous steps, and $L$ represents the length of the coordinate sequence. Our evaluation framework measures the robustness of models in complex desktop environments by calculating the Euclidean distance and the Hit Rate (i.e., whether the predicted coordinates fall within the ground truth bounding box) between the prediction $\hat{y}$ and the ground truth.

### 3.2. Data Construction

Unlike previous datasets (Hui et al., 2025; Cheng et al., 2024; Li et al., 2025), instead of directly collecting complex desktops that are difficult to annotate, we automatically generate synthetic test samples with different difficulty levels based on the collected high-fidelity single-window metadata by parametrically controlling window layout, density, occlusion ratio, and semantic similarity.

The prerequisite for constructing a high-quality synthetic benchmark is the availability of diverse and high-resolution atomic assets. To this end, we established a metadata repository containing high-fidelity screenshots, serving as a solid foundation for the subsequent benchmark construction. The data construction pipeline is illustrated in Figure 2. We

selected the widely used Windows operating system as our data collection platform to cover the software and functional scenarios most frequently encountered by users[1]. To ensure data representativeness, we identified nine core domains—including productivity tools, browsers, communication, and developer tools—based on mainstream software market share and user behavior analysis reports (Junuzovic et al., 2011; Xie et al., 2022), as detailed in Table 1.

**Screenshot Acquisition.** We developed automated scripts designed to capture application windows during natural user interaction. The scripts perform instant captures while applications are running, thereby preserving authentic window dimensions consistent with common usage habits. All screenshots were captured in a 2560×1440 desktop environment. This resolution matches the desktop background used in subsequent synthesis, avoiding window distortion caused by resizing operations and ensuring high consistency in clarity between synthetic images and real desktops. Furthermore, high-resolution captures effectively preserve fine-grained text textures and icon details within the GUI.

**Data Annotation.** We adopted a two-stage annotation strategy combining coarse and fine granularity. First, we introduced the efficient screen parsing model, OmniParser, to perform Set-of-Mark (SoM) annotation (Yang et al., 2023a) on each screenshot. OmniParser is capable of rapidly detecting the vast majority of interactive elements within the interface and generating relatively precise bounding boxes (Wan et al., 2024). However, we observed significant deviations in the accuracy of the functional description text generated by OmniParser. Therefore, in the second stage, we utilized a more powerful Vision-Language Model (VLM), QwenVL2.5-72B (Bai et al., 2025), to obtain precise textual descriptions. Specifically, we selected several representative elements from each window and fed the bounding box coordinates output by OmniParser into QwenVL2.5-72B. Leveraging Qwen's robust understanding capabilities, we generated detailed descriptions (covering functionality, appearance, etc.) for the objects within the bounding boxes, serving as the final window-level instruction-target pairs.

The resulting metadata repository comprises 585 high-resolution screenshots and 1,356 high-quality window-level instruction-target pairs. The details are shown in Figure 3.

### 3.3. WinDeskGround Benchmark

Real-world user desktop environments are characterized by high complexity and visual clutter. The parallel processing of multiple tasks inevitably leads to window stacking, dense layouts, and interference from visually similar el-

*Table 1.* The nine core domains and representative applications covered in the metadata repository.

| Domain | Example Apps |
| --- | --- |
| Productivity | Word, Notepad, Excel |
| Browsers | Google Chrome, Edge |
| Communication | Discord, Zoom, WeChat |
| Media & Ent | Spotify, Netflix |
| Utilities | 7-Zip, CCleaner |
| Developer Tools | VS Code, Docker Desktop |
| File & System | Settings, File Explorer |
| Gaming | Solitaire, Xbox App |
| Advanced Tools | PowerShell, Resource Monitor |

ements. Existing datasets, which predominantly feature single-application screenshots, fail to capture this physical veracity, resulting in a distribution shift that hinders model robustness. To bridge this gap, we propose the WinDeskGround Benchmark. By leveraging a parameterized multi-window synthesis method, we generate test samples that rigorously evaluate model adaptability across varying degrees of environmental complexity.

#### 3.3.1. PARAMETERIZED MULTI-WINDOW SYNTHESIS

We designed a synthesis algorithm to generate realistic and challenging desktop environments. The process utilizes a standard $2560 \times 1440$ Windows 11 default wallpaper as the background canvas. To simulate the chaotic nature of real-world usage, the algorithm dynamically layers windows based on three critical dimensions: Layout Density ($\mathcal{D}$), Semantic Similarity ($\mathcal{S}_{sim}$), and Occlusion Ratio ($\mathcal{O}_{ratio}$).

**Layout, Density, and Spatial Priors.** Drawing on Human-Computer Interaction (HCI) research regarding window management behaviors, our algorithm simulates realistic user arrangement preferences (Junuzovic et al., 2011; Xie et al., 2022). We introduce *Category-Position Priors* to govern the spatial distribution of windows. Unlike random placement, different application categories are constrained to specific screen regions based on usage habits

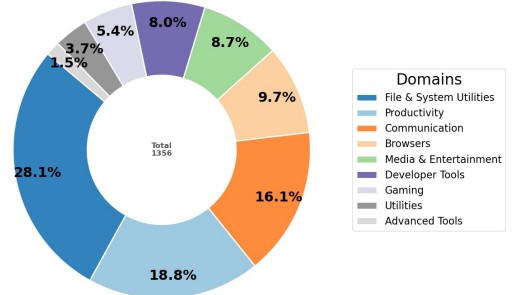

*Figure 3.* Detailed metadata distribution across domains.

---

[1]https://www.microsoft.com/en-us/store/most-popular/apps/pc

(e.g., *Communication* apps tend to reside in the periphery $[0.70, 0.95] \times [0.55, 0.85]$, while *Browsers* occupy the central viewing area). The algorithm supports varying density levels ($\mathcal{D} \in \{3, \ldots, 15\}$) and employs both cascading and tiling strategies to create visual crowding. These parameter ranges are not intended to match real-world distributions; rather, they span conditions from common usage to stress-test scenarios. Although the ranges are partially informed by prior HCI studies on window management behavior, they are primarily chosen to support controlled robustness evaluation under progressively more challenging settings.

**Semantic Similarity Injection.** Randomly stacking windows often fails to provide sufficient semantic challenges. To construct "hard negative" scenarios, we developed a similarity-based interference mechanism. First, we encode the textual descriptions of all GUI elements in our metadata repository using a pre-trained text embedding model (Reimers & Gurevych, 2019). For a given target window, we compute the cosine similarity between its target element and all other available windows, constructing a *Similar Window Sequence*. During synthesis, the algorithm prioritizes sampling "distractor" windows from this sequence. This induces high semantic ambiguity (e.g., placing a distractor containing a "Login" button next to the target "Login" button), thereby testing the model's fine-grained discrimination capabilities.

**Occlusion and Visibility Control.** Occlusion represents the most prevalent noise in multi-tasking environments. The algorithm controls the placement of distractor windows to partially obscure the target window. The occlusion intensity is quantified by the visible ratio of the target element. We enforce strict constraints on the generated samples to ensure the target remains theoretically identifiable (e.g., maintaining a visibility ratio $> 30\%$) while challenging the model to infer context from incomplete visual features. To make this constraint operational, we project the target element into global desktop coordinates, sample a desired visible ratio, and then position distractor windows with controlled overlap so that the final visibility satisfies the predefined constraint. After rendering, we recompute the actual visibility using pixel-level masks to verify correctness, thereby preventing degenerate cases in which the target becomes fully invisible. We use 30% as an empirical lower bound because preliminary manual inspection suggests that below this level both human consistency and semantic identifiability deteriorate markedly; the threshold is therefore a practical solvability trade-off rather than a theoretical optimum, and it remains configurable in our synthesis pipeline. The overall generation process is formalized in Algorithm 1.

---

**Algorithm 1** Parameterized Multi-window Desktop Synthesis

---

1: **Input:** Background $\mathcal{B}$ (2560×1440 QHD), Metadata $\mathcal{M}$, Difficulty Level $L$
2: **Output:** Synthesized Desktop Image $I$, Ground Truth $y$
3: $N_{win}, R_{occ}, N_{sim} \leftarrow$ GetParams($L$) {Get constraints from Table 2}
4: $W_{target} \leftarrow$ Sample($\mathcal{M}$)
5: $P_{target} \leftarrow$ ApplySpatialPrior($W_{target}$.category)
6: Place $W_{target}$ at $P_{target}$ on $\mathcal{B}$
7: $Q_{sim} \leftarrow$ ComputeSimilarity($W_{target}, \mathcal{M}$) {Retrieve semantic distractors}
8: $W_{distractors} \leftarrow \emptyset$
9: **for** $i = 1$ **to** $N_{win} - 1$ **do**
10:    $W_d \leftarrow$ Pop($Q_{sim}$) {Prioritize semantically similar windows}
11:    **repeat**
12:       $P_d \leftarrow$ ApplySpatialPrior($W_d$.category)
13:       $v \leftarrow$ CalculateVisibleRatio(
14:          $W_{target}, P_{target}, W_d, P_d$)
15:    **until** $v \in R_{occ}$ {Ensure occlusion constraints are met}
16:    Add $W_d$ to $W_{distractors}$
17: **end for**
18: $I \leftarrow$ Render($\mathcal{B}, W_{target}, W_{distractors}$)
19: **return** $I$, Coords($W_{target}$)

---

### 3.3.2. EVALUATION STRATEGY

Based on the proposed synthesis framework, we designed two complementary evaluation protocols to comprehensively dissect the robustness bottlenecks of current MLLMs.

**Protocol I: Single-factor Controlled Analysis.** To decouple the impact of interference factors, we employ a controlled variable approach. We evaluate performance decay by varying one parameter while keeping others constant. For instance, we fix the window density at $\mathcal{D} = 2$ and incrementally decrease the target visibility from 100% to 30%; alternatively, we fix the visibility and increase the number of semantically similar distractors. This protocol allows for the precise isolation of model weaknesses, such as sensitivity to occlusion versus confusion caused by semantic decoys. Because the remaining variables are intentionally fixed to minimal settings in this protocol, factors such as semantic similarity and visual clutter naturally produce flatter curves than occlusion; however, their effects increase once they interact with other factors in the higher-difficulty settings discussed in Section 4.2.2.

**Protocol II: Multi-level Difficulty Evaluation.** To reflect the distribution of real-world complexity, we define five dif-

ficulty levels (L1–L5). These levels integrate window count, occlusion severity, and semantic interference into a unified metric. As detailed in Table 2, scenarios range from simple "clean" desktops (L1) to extreme "stress test" environments (L5) characterized by high density ($> 10$ windows), severe occlusion (up to 70% blocked), and intense semantic confusion. This graded evaluation enables us to delineate the "capability boundary" of GUI grounding models.

# 4. Experiments and Results

## 4.1. Baselines and Evaluation Metric

**Models and Datasets.** To comprehensively assess the robustness of existing techniques in complex desktop environments, we conduct experiments on our **WinDeskGround** benchmark. We select five representative open-source GUI grounding models as baselines: **SeeClick** (Cheng et al., 2024), a vision-only model fine-tuned on the Qwen-VL (Bai et al., 2025) architecture known for its high precision on mobile and web interfaces; **OS-Atlas-7B-Base** (Wu et al., 2024), a foundational GUI grounding model trained on a large-scale, cross-platform action dataset designed for general-purpose computer control; **UGround-7B** (Gou et al., 2025), a universal grounding model emphasizing cross-platform generalization; **UI-TARS-1.5-7B** (Wang et al., 2025), a state-of-the-art GUI agent model integrating large-scale instruction tuning with reinforcement learning; and **InfiGUI-G1-7B** (Liu et al., 2025), which utilizes adaptive exploration policy optimization to advance GUI grounding performance. These models are evaluated across the difficulty levels (**L1** to **L5**) of WinDeskGround, specifically testing their adaptability to varying degrees of window density, occlusion ratios, and semantic interference.

**Evaluation Metric.** We employ **Click Accuracy** as the primary metric to quantify model performance. This metric measures whether the coordinate point predicted by the model falls within the effective clickable region of the target. Formally, for the $i$-th test sample, let $\hat{y}_i$ denote the predicted coordinate point and $\mathcal{B}_i$ represent the ground truth bounding box. A prediction is considered a "Hit" if and only if the point lies inside the bounding box. The average Click Accuracy over the dataset is defined as:

$$\text{Accuracy} = \frac{1}{N} \sum_{i=1}^{N} \mathbb{I}(\hat{y}_i \in \mathcal{B}_i) \qquad (2)$$

where $N$ is the number of samples and $\mathbb{I}(\cdot)$ is the indicator function. This metric directly reflects the agent's ability to correctly trigger the target widget in real-world operations.

## 4.2. Results

### 4.2.1. SINGLE-FACTOR CONTROLLED ANALYSIS

To dissect the specific impact of environmental interference on GUI grounding performance, we conducted controlled variable experiments on the WinDeskGround benchmark, as illustrated in Figure 4. The results reveal distinct behavioral patterns across visual clutter, occlusion, and semantic interference dimensions.

Regarding **visual clutter** (Figure 4a), contrary to the expectation of performance degradation, top-tier agents exhibit remarkable stability. **UGround**, **UI-TARS**, and the newly introduced **InfiGUI** maintain high accuracy levels (ranging from 70% to 80%) even as the window count increases to 12. This demonstrates their resilience to dense layouts. **OS-Atlas** maintains a mid-tier performance ($\sim$50%), while **SeeClick** struggles significantly, hovering below 20%. This distinct stratification indicates that the dynamic high-resolution support and adaptive policies in advanced architectures successfully mitigate the noise introduced by window stacking, whereas models limited to fixed low resolutions fail to isolate targets in crowded environments.

**Occlusion** emerges as the most critical bottleneck for all models (Figure 4b). We observe a precipitous drop in accuracy across all architectures as target visibility decreases. While **UGround**, **UI-TARS**, and **InfiGUI** start with strong performance ($>75\%$) at full visibility, their accuracy collapses to below 20%—converging with weaker models—when visibility falls to the 30–50% range. This finding underscores a shared limitation: current MLLMs rely heavily on complete visual features for matching and lack the reasoning capability to infer whole objects from severely fragmented visual cues.

In terms of **semantic interference** (Figure 4c), the results indicate unexpected robustness. Instead of a linear decline, the performance curves remain relatively flat across increasing similarity levels (1–5). This stability is likely attributed to the layering logic where distractor windows are positioned behind the target, allowing models to prioritize foreground elements. We intentionally adopt this conservative design to decouple semantic interference from occlusion effects and preserve interpretability of the factor analysis, although it may underestimate the full difficulty of semantic ambiguity in real-world scenarios. **UGround** consistently leads the benchmark, followed closely by **UI-TARS** and **InfiGUI**, which exhibit nearly identical performance trajectories. This suggests that provided the visual features are clearly visible, the language understanding capabilities of these top-tier agents are sufficiently robust to distinguish targets from "hard negative" distractors (e.g., distinguishing a target "Login" button from a semantically identical decoy). However, in the subsequent comprehensive evaluation, the

*Table 2.* **Definition of Difficulty Levels in WinDeskGround Benchmark.** We categorize desktop complexity into five levels based on three key dimensions: Window Count ($N_{win}$), Target Visible Ratio ($V_{target}$), and the level of Semantically Similar Distractors ($L_{sim}$).

| Level | Window Count ($N_{win}$) | Visible Ratio ($V_{target}$) | Sim. Distractors ($L_{sim}$) | Complexity | Description |
|---|---|---|---|---|---|
| **L1** | $2 - 4$ | $1.00 - 1.00$ | 1 | Low | Simple desktop. |
| **L2** | $4 - 6$ | $0.80 - 0.90$ | 2 | Moderate | Common usage. |
| **L3** | $6 - 9$ | $0.70 - 0.80$ | 3 | High | Busy workflow. |
| **L4** | $8 - 12$ | $0.50 - 0.70$ | 4 | Very High | Dense layout. |
| **L5** | $10 - 15$ | $0.30 - 0.50$ | 5 | Extreme | Chaotic. |

compounding effect of distractors combined with occlusion further escalates the difficulty, resulting in overall performance falling below that of these single-factor experiments.

### 4.2.2. MULTI-LEVEL DIFFICULTY EVALUATION

To rigorously quantify the impact of environmental complexity on model robustness, we evaluated agent performance across a spectrum of difficulty levels, ranging from idealized single windows to highly chaotic desktops. Figure 5 illustrates the performance trajectories as difficulty escalates.

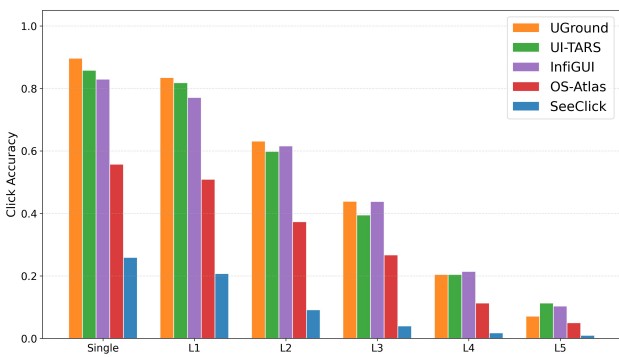

*Figure 5.* Multi-level Difficulty Evaluation. Comparison of Click Accuracy across varying difficulty levels. While all models perform well in the Single Window setting, InfiGUI demonstrates superior resilience in the mid-to-high complexity range (L2–L4).

In the baseline **Single Window** setting, distinct tiers of capability are apparent. The top-tier agents—**UGround**, **UI-TARS**, and **InfiGUI**—demonstrate dominant performance, all achieving accuracies exceeding 80%. **OS-Atlas** occupies the middle tier (~55%), while **SeeClick** struggles significantly, starting with a low accuracy of ~25%. This suggests that the low-resolution model is insufficient even in idealized scenarios without advanced reasoning.

As environmental complexity increases (**L1 to L4**), performance differentiation becomes more pronounced. While all models experience decay, **InfiGUI** exhibits resilience. Notably, in the mid-difficulty range (L2 and L3), InfiGUI matches or even slightly surpasses the performance of UI-TARS and UGround (e.g., at L2, InfiGUI achieves 61.58%

compared to UI-TARS's 59.81%). By level **L4**, InfiGUI emerges as the most robust model (21.39%), suggesting that its adaptive exploration policy effectively mitigates the confusion caused by moderate occlusion and clutter.

In the extreme stress test environment of **L5**, characterized by the compounding effects of dense occlusion and high semantic interference, all models converge to a near-failure state. However, a divergence exists: **UI-TARS** and **InfiGUI** maintain a residual accuracy of approximately 10–11%, whereas **UGround** suffers a sharper decline to 7.08%. These results confirm that while state-of-the-art GUI agents handle moderate clutter, the combination of severe occlusion and semantic ambiguity remains a bottleneck, though agents with reinforcement learning (UI-TARS) or adaptive policies (InfiGUI) show slightly better survivability.

### 4.2.3. PERFORMANCE BY APPLICATION CATEGORY

The analysis of L3 samples (Table 3) highlights performance variations across domains. Models generally achieve peak accuracy in Media and Gaming (~50%), likely due to their distinct, icon-heavy interfaces.

A key improvement is observed in the **Advanced Tools** category, which is typically challenging for agents. While baselines like UGround and UI-TARS plateau around 14%, **InfiGUI** reaches 28.57%, suggesting superior adaptability to complex professional software.

Overall, **InfiGUI** achieves the highest accuracy of 44.89%, specifically leading in information-dense categories such as **Browsers** (56.06%) and **File & System** (41.21%). **UGround** remains competitive in **Communication** tools (52.75%), indicating complementary strengths across architectures. In contrast, **SeeClick** yields ineffective results across all domains (<5%).

## 5. Case Study: Failure Analysis

To better understand the failure modes of state-of-the-art agents, we visualize two representative error cases of the **UGround** model in Figure 6.

**Semantic Interference** (Figure 6a). The user instruction is

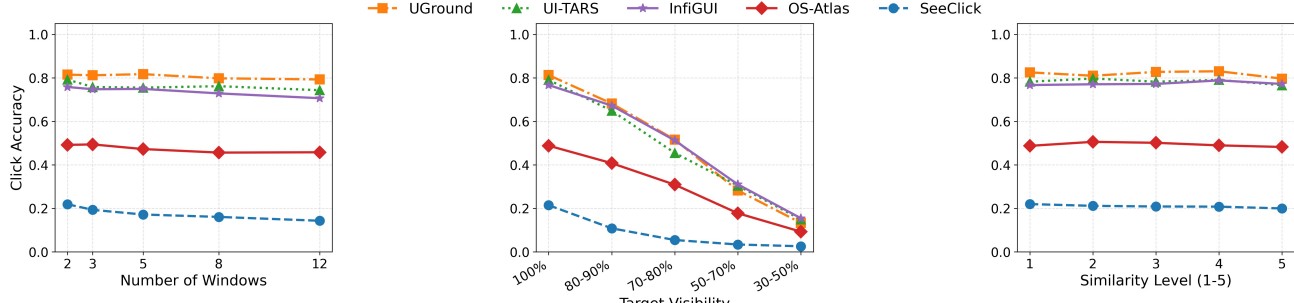

*Figure 4.* Robustness analysis under controlled variables. We evaluate the models' Click Accuracy against varying levels of (a) Visual Clutter, (b) Occlusion, and (c) Semantic Interference.

*Table 3.* Performance by Application Category (L3 Samples). We report the Click Accuracy (%) for each model across nine core domains. InfiGUI demonstrates balanced performance and leads in the challenging Advanced Tools category.

| Model | Adv. Tools | Browsers | Comm. | Dev. Tools | File & Sys. | Gaming | Media | Productivity | Utilities | Overall |
|---|---|---|---|---|---|---|---|---|---|---|
| SeeClick | 0.00 | 5.30 | 4.13 | 0.93 | 2.10 | 15.07 | 7.63 | 3.14 | 2.00 | 4.48 |
| OS-Atlas | 4.76 | 34.85 | 25.23 | 29.63 | 17.32 | 41.10 | 42.37 | 25.88 | 32.00 | 28.13 |
| UI-TARS | 14.29 | 42.42 | 42.20 | 39.81 | 32.28 | 53.42 | 46.61 | 38.43 | **52.00** | 40.16 |
| UGround | 14.29 | 50.00 | **52.75** | 40.74 | 37.27 | 49.32 | **55.08** | **40.00** | 44.00 | 42.61 |
| InfiGUI | **28.57** | **56.06** | 38.99 | **42.59** | **41.21** | **54.79** | 53.39 | 38.43 | 50.00 | **44.89** |

to click the "Like" button (thumbs-up icon) within the video player interface. The target is prominent in the foreground. However, the model is distracted by a semantically identical icon located in the background content (marked by the red dot). Although visually similar, the background element is contextually irrelevant to the primary interaction flow. This indicates that while UGround correctly identifies the semantic category of elements, it remains highly susceptible to distraction by similar elements when relying on relatively brief reference expressions.

**Partial Occlusion** (Figure 6b). In this scenario, the target is the "Paste" option within a menu. Despite being partially occluded by an overlapping window, the target remains recognizable to humans through multimodal cues—the "Paste" text is partially legible, and the icon is half-visible. The model, however, ignores this partially obscured target and incorrectly selects a fully visible, visually similar icon elsewhere on the screen. This reveals a visibility bias: current MLLMs tend to over-rely on complete visual features and struggle to reason about object permanence or infer whole entities from fragmented visual and textual clues.

## 6. Human Validation of Dataset Quality

To ensure the reliability of *WinDeskGround*, we conducted a human verification study assessing both instruction clarity and annotation precision. We randomly sampled 100 instances from the generated dataset. Three independent human evaluators were tasked with binary validation based on two criteria: (1) **Instruction Validity**: Does the tex-

*Table 4.* Validation quality across difficulty levels.

| Level | BBox Accuracy | Target Clickable | Multiple Valid Targets |
|---|---|---|---|
| Overall | 93.20% | 94.40% | 9.64% |
| L1 | 99.00% | 99.20% | 4.60% |
| L2 | 98.40% | 99.40% | 7.20% |
| L3 | 97.40% | 98.00% | 9.40% |
| L4 | 91.60% | 92.40% | 11.60% |
| L5 | 79.60% | 83.00% | 15.40% |

tual instruction uniquely and accurately describe the target element? and (2) **Spatial Accuracy**: Is the ground truth bounding box precise? The evaluation yielded an average approval rate of **85%** for instruction validity and **99%** for spatial accuracy. Notably, the 85% instruction-validity figure is measured under strict criteria requiring unique and unambiguous interpretation, so many rejected cases reflect boundary ambiguity rather than incorrect annotations. The inter-annotator agreement (Fleiss' Kappa) was 0.87, indicating high consistency and confirming that our automated synthesis pipeline produces high-quality data suitable for rigorous benchmarking. To provide a more comprehensive view, we further analyze quality across difficulty levels in Table 4; low-difficulty samples (L1–L3) remain very clean ($\geq 97\%$), whereas higher levels (L4–L5) introduce more ambiguity as a natural consequence of increasingly realistic and challenging GUI scenes.

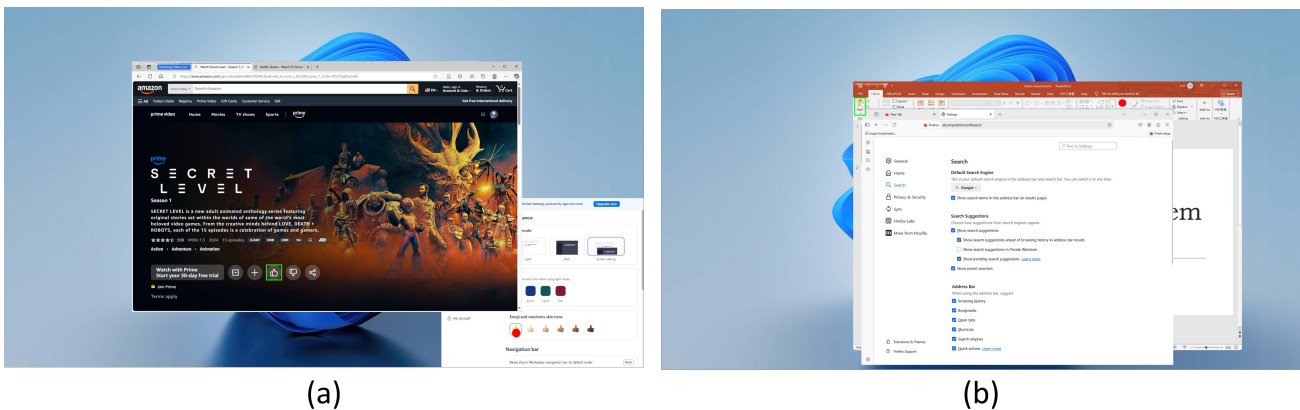

(a)                                                          (b)

*Figure 6.* Qualitative analysis of UGround failure cases. The **Green Box** denotes the ground truth target, while the **Red Dot** indicates the model's incorrect prediction. (a) **Semantic Interference:** The model fails to distinguish the primary "Like" button from a background distractor. (b) **Occlusion:** The model skips the partially occluded target "Paste" option in favor of a fully visible but incorrect icon.

# 7. Realism and Synthetic-to-Real Gap Analysis

While our benchmark relies on synthetic composition, it is constructed from real desktop screenshots and annotations, making it a hybrid dataset rather than fully synthetic. To assess the realism of the generated scenes, we conduct a perceptual evaluation using vision-language models as judges, where each synthesized multi-window scene is independently evaluated for its realism.

This evaluation yields an average realism score of 86.68%, indicating that the synthesized scenes are perceived as highly realistic. At the same time, our goal is not to replace real-world benchmarks, but to provide a controllable testbed for analyzing specific robustness factors such as occlusion and layout complexity; future work will therefore investigate how performance on WinDeskGround correlates with downstream computer-use benchmarks collected from real environments.

# 8. Discussion and Future Work

**Limitations.** While WinDeskGround offers a structured robustness evaluation, it relies on static snapshots, which cannot fully capture the temporal dynamics of real-world usage (e.g., hover states, focus switching). Furthermore, the current focus on single-step grounding excludes multi-step reasoning, such as manipulating windows to reveal occluded targets, thereby limiting the assessment of long-horizon planning capabilities. More specifically, WinDeskGround centers on single-step grounding and does not model interaction dynamics such as window switching or focus changes; while this simplification helps isolate perception ability, it does not fully capture real-world interaction constraints. In addition, although our synthesis pipeline improves controllability, it may introduce structural biases such as layering assumptions or spatial priors. Preliminary ablations suggest

that these effects are limited, but future work should examine them more systematically and incorporate more diverse layout distributions.

**Future Work.** Future research should extend evaluation to interactive dynamic environments. To mitigate occlusion, integrating Multimodal Retrieval-Augmented Generation (RAG) offers a path to recover blocked context from historical snapshots without disturbing the layout. Additionally, combining visual inputs with system Accessibility Trees (hybrid modal augmentation) could significantly enhance robustness in visually compromised scenarios, accelerating the transition from prototypes to reliable real-world tools.

# 9. Conclusion

In this paper, we address the critical robustness challenges faced by MLLMs in desktop GUI automation by introducing WinDeskGround, the first comprehensive benchmark tailored for evaluating grounding performance in complex multi-window environments. By leveraging high-fidelity metadata and a scalable parametric synthesis framework, we successfully simulate realistic scenarios characterized by window stacking, dense layouts, and semantic interference, effectively bridging the gap left by overly idealized existing datasets. Extensive evaluations on WinDeskGround reveal a distinct reality gap: while SOTA models excel in simplified settings, their performance degrades significantly when facing the complexity of real-world desktops. Our analysis identifies vulnerability to occlusion and sensitivity to semantic interference as the primary obstacles to reliable deployment. We envision WinDeskGround as a standard stress test for future research, driving the transition of GUI agents from idealized laboratory environments to trustworthy and robust real-world applications.

## Impact Statement

This paper introduces WinDeskGround, a benchmark for evaluating the robustness of multimodal models in complex multi-window desktop GUI grounding, with the goal of improving evaluation methodologies and the reliability of machine learning systems for computer use. All screenshots are collected from controlled environments and reviewed to avoid personal or sensitive information, and the benchmark relies on synthetic multi-window compositions to further reduce privacy risks. However, the selection of applications and the use of automated annotation pipelines based on existing vision–language models may introduce biases that affect representativeness. We encourage transparent documentation and cautious interpretation of results when using this benchmark. More broadly, advances in GUI automation may benefit accessibility and productivity while also raising considerations around user autonomy and responsible deployment; WinDeskGround is intended as an evaluation resource and should be used in a manner that respects fairness, accountability, and consent.

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
