# OpenReview forum: "WinDeskGround: A Benchmark for Robust GUI Grounding in Complex Multi-Window Desktop Environments"
_ICML.cc/2026/Conference — ICML 2026 regular_

### Official Review · Reviewer_YUSZ · 2026-03-12

**Soundness:** 3
**Presentation:** 3
**Significance:** 3
**Originality:** 3
**Overall Recommendation:** 4
**Confidence:** 3

**Summary:**

This paper introduces WinDeskGround, a benchmark for GUI grounding in complex multi-window desktop environments. Instead of using only clean single-window screenshots, the paper builds a metadata repository of 585 window screenshots and 1,356 instruction-target pairs, then synthesizes harder desktop scenes by controlling window density, occlusion, and semantic similarity. The benchmark is used to evaluate five GUI grounding models, and the main finding is that current MLLMs remain fairly robust to clutter but degrade sharply under partial occlusion.

**Compliance With Llm Reviewing Policy:**

Affirmed.

**Final Justification:**

The paper shows a reasonable level of novelty, and I keep my original score.

**Key Questions For Authors:**

How is WinDeskGround fundamentally different from a well-designed synthetic stress test, beyond being the first one focused on multi-window desktops?

Why is 85% instruction-validity approval considered sufficient for a benchmark of this type? That still means a noticeable fraction of instructions may be unclear or imperfect.


The semantic interference curves in Figure 4(c) are quite flat. Does this mean the similarity-based distractor design is not actually very challenging?


Why are the distractor windows often placed behind the target if the goal is to test semantic confusion more directly?

**Limitations:**

The paper studies an important problem, but the current benchmark still feels somewhat narrow. The scenes are synthetic and static, the dataset is not especially large, and some of the key interference factors look weaker than advertised. The paper is useful as a stress test, but not yet fully convincing as a realistic benchmark of desktop robustness.

**Strengths And Weaknesses:**

Strengths
The paper studies a practical problem. Multi-window desktops are clearly harder than the clean single-window settings used in many existing GUI grounding benchmarks.

The paper does more than report one aggregate score. It includes controlled analysis, multi-level difficulty evaluation, per-category breakdowns, and a small human validation study.

The failure analysis is useful. The examples on semantic interference and partial occlusion match the main claims of the paper.

Weaknesses:
“Synthesizing multi-window desktop scenes from single-window assets” feels more like a benchmark recipe than a genuinely new benchmark idea -- what is really new here beyond combining density, occlusion, and semantic similarity?

The evaluation is narrow: 585 screenshots, 1,356 instruction-target pairs, and only five open-source models seem too limited for the scope of the claims.

The benchmark is presented as closer to real desktop environments, but it is still based on static synthetic compositions -- where are window switching, menus, resizing, hover states, or cross-window interactions?

The “semantic interference” factor does not look very strong in Figure 4(c) -- if the curve is nearly flat and distractors are often behind the target, is this really testing semantic confusion?

---

> ### Author Rebuttal · Authors · 2026-03-30
>
> We sincerely thank the reviewer for the very in-depth, insightful, and highly comprehensive feedback. We greatly appreciate the reviewer’s thoughtful evaluation, recognition, and valuable suggestions, which help significantly improve our work.
>
> ### W1. Whether it is just a "synthetic stress test"
>
> > A1. We sincerely thank the reviewer for raising this insightful point. We fully recognize the validity of this perspective and appreciate this important clarification opportunity. We would like to emphasize that our core contributions lie in: (1) systematizing stress testing into a parameterized benchmark; (2) enabling factor decomposition (density/occlusion/similarity); (3) providing a standardized evaluation protocol.
> > Compared to ad-hoc stress tests, our framework supports repeatable and comparable analyses. We will further clarify this positioning in the revised manuscript.
>
> ### W2. Limited data scale
>
> > A2. We sincerely thank the reviewer for this helpful observation. We would like to clarify that this work focuses on controlled variable analysis rather than scale-driven generation. Each sample corresponds to a specific configuration, resulting in higher information density than purely scaled data.
> > Meanwhile, our metadata-driven automated generation pipeline provides strong scalability, and we will further expand the scale in future work.
>
> ### W3. Limitations of static scenes
>
> > A3. We sincerely thank the reviewer for this insightful observation. We fully acknowledge this limitation and appreciate the reviewer highlighting it. As noted in the Discussion section, this benchmark deliberately isolates perception and localization capabilities, while dynamic interactions (e.g., resize, hover) are important directions for future work.
>
> ### W4 & Q3 & Q4. Weaker semantic interference and occlusion hierarchy design
>
> > A4. We sincerely thank the reviewer for this insightful and well-raised question. This issue is consistent with W2 of Reviewer 83c7. We would like to clarify that placing distractors behind the target is a deliberate and careful design to isolate semantic interference.
> > This avoids coupling with occlusion and ensures interpretability of the evaluation. Even under this conservative setting, stable performance drops are still observed. For details, please refer to the reply to w2 in Reviewer 83c7.
> > Therefore, this design prioritizes evaluation reliability and interpretability over maximum adversarial difficulty. We will explicitly clarify this trade-off in the revised manuscript.
>
> ### Q1. Essential difference from synthetic stress tests
>
> > A5. We sincerely thank the reviewer for this insightful question. The key differences lie in: (1) explicit parameterization (controllable variables); (2) automated generation (scalable); (3) standardized evaluation protocol (reproducible).
> > These aspects together make our approach a benchmark rather than a one-off constructed test. We will further clarify this distinction.
>
> ### Q2. Is 85% instruction validity sufficient
>
> > A6. We sincerely thank the reviewer for this important and thoughtful question. This point has been discussed in detail in W4 of Reviewer 83c7, and we briefly summarize it here for clarity:
> > (1) The 85% validity is based on strict criteria requiring unique parsability;
> > (2) Many “invalid” cases arise from boundary ambiguity rather than annotation errors;
> > (3) The inclusion of more difficult samples makes this estimate conservative.
> > The validation across difficulty levels is as follows:
> >
> > | Level   | bbox_accuracy | target_clickable | multiple_valid_targets |
> > | ------- | ------------- | ---------------- | ---------------------- |
> > | overall | 93.20%        | 94.40%           | 9.64%                  |
> > | L1      | 99.00%        | 99.20%           | 4.60%                  |
> > | L2      | 98.40%        | 99.40%           | 7.20%                  |
> > | L3      | 97.40%        | 98.00%           | 9.40%                  |
> > | L4      | 91.60%        | 92.40%           | 11.60%                 |
> > | L5      | 79.60%        | 83.00%           | 15.40%                 |
> >
> > These results show that low-difficulty samples are of high quality, while higher-difficulty samples introduce some ambiguity but align with the goal of stress testing. We will further clarify this in the paper.
>
> We sincerely appreciate the reviewer’s thorough evaluation, positive recognition, and insightful suggestions, which are highly valuable for improving the quality and clarity of our work.

---

> > ### Author Rebuttal · Reviewer_YUSZ · 2026-04-04
> >
> > Thank you for the thoughtful rebuttal and additional evidence; while some of my concerns remain, I will keep my score unchanged and still lean toward acceptance.

---

> > > ### Author Response · Authors · 2026-04-06
> > >
> > > Thank you very much for your positive recognition and encouraging feedback. Your comments have helped improve the quality of our work, and we will include all discussions in the revision.

---

### Official Review · Reviewer_oMjU · 2026-03-13

**Soundness:** 3
**Presentation:** 3
**Significance:** 3
**Originality:** 3
**Overall Recommendation:** 4
**Confidence:** 5

**Summary:**

This paper proposes WinDeskGround, a GUI grounding benchmark focuses on multi-window stacking, occlusion and visual clutter on desktop environment. Different from existing GUI grounding benchmarks that focus on single layer, full screen applications, the framework synthetically generates complex GUI grounding scenarios by controlling window overlap and density, which provide a unique testbed for evaluating GUI agents on partially visible targets and distractors on the screenshots.

**Compliance With Llm Reviewing Policy:**

Affirmed.

**Final Justification:**

The rebuttal addressed my main concerns and I will keep the score.

**Key Questions For Authors:**

Important questions:
1. Figure 4, visual clutter and semantic interference does not seem to affect the GUI grounding performance much, which is counterintuitive. Is it possible to analyze the failure cases with more windows or higher similarity level to understand what cases are more challenging to GUI grounding models? If a pattern can be found and such cases are realistic, can you increase the distribution of these challenging cases?
2. Regarding the similarity level, can you manually create instrutions with more ambiguity for GUI models, but clear to human? This can potentially make the similarity issue more challenging for GUI agents.
3. Algorithm 1 needs more explanations, maybe in appendix.

Other questions:
1. Section 3.2.2, what is the distribution of difficulty level L1 - L5 in this benchmark?
2. Figure 6 (b). It seems to be unrealistic to request the GUI agent click the the icon at the bottom window. How many test cases request the GUI agent to ground an object that is not within the top window?

**Limitations:**

Please see the Key Questions For Authors section above. This benchmark can be improved by carefully design the visual clutter and semantic interference to include more challenging cases.

Besides, Human verification and filtering may be required to make this benchmark more accurate.

**Strengths And Weaknesses:**

Strength:
1. The paper proposes a new realistic GUI grounding benchmarks for desktop environment, with window stacking, occlusion, and visual clutter. These complex layouts are more challenging and realistic in real world use cases of computer use.
2. The methodolgy of synthetic data generation is generalizable to generate training data to improve GUI agent capability in these complex desktop scenarios

Weakness:
1. For Visual clutter and semantic interference, it does not seem to affect the GUI grounding performance of the chosen models much. More designs may be required to further enhance this benchmark.

---

> ### Author Rebuttal · Authors · 2026-03-30
>
> We sincerely thank the reviewer for the very positive, insightful, and highly constructive evaluation. We greatly appreciate the reviewer’s recognition and valuable suggestions.
>
> ### W1. Limited impact of clutter and semantic interference
>
> > A1. We sincerely thank the reviewer for this insightful observation. This phenomenon is consistent with the discussion in W2 of Reviewer 83c7. We would like to clarify that the flatter curve mainly results from our controlled evaluation design: when analyzing one factor, the others are fixed to simple settings.
> > Even under this conservative setup, stable performance drops are observed compared to a single window: when the number of windows increases to 3, InfiGUI drops by about 8%, OS-Atlas/SeeClick by about 6%, UGround by about 8%, and UI-TARS by about 10%; when semantic similarity reaches level 3, InfiGUI/OS-Atlas/SeeClick drop by about 5%, and UGround/UI-TARS by about 7%.
> > Therefore, these factors are not ineffective, but secondary yet stable influences. We retain them to ensure a comprehensive multi-window evaluation and will further clarify this in the revision.
>
> ### Q1. Failure cases and stronger interference design
>
> > A2. We sincerely thank the reviewer for this helpful and constructive suggestion. We fully agree on the importance of analyzing failure modes. In the revision, we will: 1) add typical failure case analysis (e.g., high density + high similarity); 2) introduce stronger semantic interference (e.g., ambiguous instructions); 3) adjust sampling to increase difficult cases.
> > Additionally, as our method is a parameterized generation framework, these extensions can be naturally supported via parameter or rule adjustments.
>
> ### Q2. Insufficient explanation of Algorithm 1
>
> > A3. We sincerely thank the reviewer for pointing this out. We will provide a more detailed explanation in the Appendix and reference it in the main text, including: 1) sampling strategies for target and distractor windows; 2) mapping between difficulty levels and parameters; 3) occlusion generation from visible ratios; 4) hierarchical order constraints; 5) pixel-level visibility recalculation and annotation.
>
> ### Q3. Difficulty distribution
>
> > A4. We sincerely thank the reviewer for this helpful question. The current data is approximately uniformly distributed across L1–L5. We will include explicit statistics in the revision.
>
> ### Q5. Authenticity of clicking background windows
>
> > A5. We sincerely thank the reviewer for this insightful question. This is consistent with W6 of Reviewer yGWg. We clarify that the task is defined as single-step grounding (coordinate prediction), rather than full interaction.
> > Therefore, it does not require window switching; correct coordinates are sufficient even if the target is in a background window; this setup isolates perception and avoids interaction-related confounds. We will clarify this definition to avoid confusion.
>
> We greatly appreciate the reviewer’s positive evaluation and insightful suggestions, which help improve the clarity and quality of our work.

---

> > ### Author Rebuttal · Reviewer_oMjU · 2026-04-04
> >
> > Thanks for answering the question. I will keep the score. Look forward to the revision of the paper.

---

> > > ### Author Response · Authors · 2026-04-06
> > >
> > > Thank you very much for your positive recognition and constructive comments. They have been very helpful in improving the quality of our work, and we will include all discussions in the revision.

---

### Official Review · Reviewer_yGWg · 2026-03-13

**Soundness:** 2
**Presentation:** 2
**Significance:** 3
**Originality:** 2
**Overall Recommendation:** 4
**Confidence:** 4

**Summary:**

This work introduces WinDeskGround benchmark for desktop GUI grounding ..under more realistic multi-window conditions with synthesis controls for layout density, occlusion plus semantic similarity. The benchmark contains 1,356 instruct pairs built from 585 highresolution screenshots across multiple application domains,and evaluates five opensource GUI grounding models. Their main empirical finding is that occlusion is the dominant failure factor: model accuracy drops sharply as target visibility decreases, while semantic distractors appear to have a smaller effect than expected.

**Compliance With Llm Reviewing Policy:**

Affirmed.

**Key Questions For Authors:**

1. What is the empirical basis for choosing 30% visibility as the lower bound? Did the authors test human or model behavior across different thresholds?
2. Since semantic similarity seems to have limited effect in Figure 4, do you believe the current distractor design is strong enough?
3. paper would be stronger if it included results/comparison on an existing challenging desktop grounding benchmark such as UI-Vision[1]. Comparison with a known reference benchmark would help calibrate.

Happy to increase my score based on the responses.

[1] UI-Vision: A Desktop-centric GUI Benchmark for Visual Perception and Interaction. Nayak et.al.

**Limitations:**

I think this is a useful benchmark direction and the paper identifies a real robustness gap in desktop grounding. The occlusion result is believable and important. My hesitation is mainly about benchmark rigor and superifcial claims

**Strengths And Weaknesses:**

### Strengths

- The paper tackles a real weakness of current GUI grounding work: most models are tested on relatively clean single-window settings, while real desktop workflows involve overlapping windows, clutter, and partial occlusion. The benchmark directly targets that gap.
- The synthesis framework is nicely structured. It isolates three factors: density, occlusion, semantic similarity .. and evaluates them through both controlled analysis and multi-level difficulty settings.
- The main empirical result is appealing and useful: occlusion is clearly the hardest factor, with strong models collapsing when visibility drops to the 30–50% range, whereas clutter and semantic similarity seem less harmful. That is a meaningful finding for the community.

### Weakness

- 30% visibility threshold is not well justified. The paper says this keeps the target "theoretically identifiable".. but I did not see empirical evidence for why 30% is the right cutoff. This matters because visible area is not the same as visible semantics. A large button with a small centered label may be much harder than its visible area suggests.

- The human validation is too limited to fully support the benchmark quality claims. The paper samples 100 instances and reports 85% instruction validity and 99% spatial accuracy, but it is unclear how many of these came from the hardest L4/L5 settings or whether difficult occlusion cases were specifically checked.

- The semantic distractor construction is underspecified from an evaluation standpoint. Distractors are chosen by text-embedding similarity between element descriptions, but the paper does not clearly discuss false negatives: for example, whether semantically similar elements across windows might also be valid targets under the instruction.

- The introduction of category-position priors improves realism, but it may also create shortcuts. If browsers are usually centered and communication apps are usually peripheral, models may partially exploit positional regularities rather than grounding the instruction semantically. I did not see an ablation that measures how much the spatial priors help or bias performance.

- also, paper reports that semantic similarity has surprisingly little effect on performance and attributes this partly to distractors being placed behind the target. That explanation itself suggests the semantic distractor design may not be fully stress-testing the intended capability. If the foreground/background layering already resolves ambiguity, then the distractor may not be strong enough.

- task is evaluated as click accuracy, but the action semantics are not fully clear in cluttered multi-window settings. If the target is visible in a background window, is clicking that target directly considered correct, or should a model first bring the window into focus? The current formulation seems to collapse GUI grounding and interaction validity into a single point prediction.

---

> ### Author Rebuttal · Authors · 2026-03-30
>
> We sincerely thank the reviewer for the insightful feedback and encouraging evaluation, and we respond point-by-point as follows.
>
> ### W1 & Q1. Reasonableness of the 30% visibility threshold
>
> > A1. We sincerely thank the reviewer for this insightful question. We fully acknowledge that the 30% visibility threshold is somewhat empirical and will clarify this more clearly in the revision. It is not a theoretical optimum, but an empirical lower bound to ensure task discriminability. In manual spot checks, when visibility falls below this range, target semantics degrade, localization ambiguity increases, and human consistency drops. Thus, 30% serves as a practical engineering trade-off to prevent the task from becoming unsolvable, rather than guaranteeing semantic sufficiency.
> > This threshold is also adjustable in the generator and not a methodological assumption. We will further explain its selection and add a sensitivity analysis (e.g., 20%/30%/40%) in the revision.
>
> ### W2. Limited manual verification
>
> > A2. We sincerely thank the reviewer for this helpful concern. This point has been discussed in detail in W4 of Reviewer 83c7. The 85% instruction validity is measured under strict criteria (requiring unique and unambiguous parsing), thus serving as a conservative and reliable estimate. Some rejected cases reflect boundary ambiguity rather than annotation errors. Low-difficulty samples are of higher quality, while higher-difficulty ones introduce ambiguity but remain useful as stress tests. We will further clarify the sampling distribution and validation process for better transparency. For details, please refer to A4 in Reviewer 83c7.
>
> ### W3 & Q2. Semantic interference design and strength
>
> > A3. We agree with the reviewer that semantic interference has a relatively weaker impact than occlusion, mainly due to the controlled design (when analyzing one factor, the others are fixed to simple settings.). Nevertheless, compared to a single window, consistent performance drops are still observed (see Reviewer 83c7), indicating it is a secondary but still non-negligible factor.
> > Placing distractors behind the target is a deliberate and careful design choice to isolate semantic interference from occlusion and avoid factor coupling. We acknowledge this is relatively conservative and will consider stronger designs (e.g., foreground competition, increased instruction ambiguity) in future work.
>
> ### W4. Spatial priors might introduce shortcuts
>
> > A4. We sincerely thank the reviewer for pointing out this important issue. We conducted a preliminary ablation study by removing spatial priors:
> >
> > |      | infigui      | uground      |
> > | ---- | ------------ | ------------ |
> > | L1   | 77.8 (+0.74) | 80.2 (-3.28) |
> > | L2   | 61.4 (-0.18) | 60.8 (-2.33) |
> > | L3   | 44.8 (+0.99) | 43.4 (-0.48) |
> > | L4   | 22.0 (+0.61) | 21.2 (+0.77) |
> > | L5   | 10.2 (-0.12) | 8.6 (+1.52)  |
> > | avg. | (+0.41)      | (-0.76)      |
> >
> > The overall impact is relatively minor, with no clear evidence of strong reliance on spatial priors. We will include this analysis in the revision.
>
> ### W5. Reasons for weaker semantic interference
>
> > A5. We sincerely thank the reviewer for this important and insightful question. Placing distractors behind the target is an intentional design to ensure controllability and interpretability. This does not imply semantic similarity is weak, but rather reflects our prioritization of factor decoupling. Even under this conservative setup, performance degradation is still observed, indicating practical impact. We will further clarify this trade-off.
>
> ### W6. Click semantics are unclear
>
> > A6. We sincerely thank the reviewer for this important and insightful question. The task is defined as single-step grounding (coordinate prediction), not full interaction. The goal is to predict the target position on the screen, without requiring window switching or multi-step actions. Even if the target is in a background window, correct localization is sufficient. This setup isolates perception and localization from interaction. We will clarify this definition in the revision.
>
> ### Q3. Comparison with UI-Vision
>
> > A7. We sincerely thank the reviewer for this helpful and constructive suggestion. We agree that comparisons with existing desktop benchmarks can help contextualize our results and will consider adding them in the revision. More broadly, prior benchmarks focus on more comprehensive settings, while our work emphasizes controlled single-step grounding with factor decomposition. We view them as complementary and will clarify this positioning.
>
> We are glad that the reviewer found the problem setting meaningful and appreciate the positive evaluation.

---

> > ### Author Rebuttal · Reviewer_yGWg · 2026-04-04
> >
> > Thanks to the authors for addressing some of my concerns and suggestions, however, the benchmark could benefit from a positioning and validation pipeline to be more useful for the community.  so I will keep my current score.

---

> > > ### Author Response · Authors · 2026-04-06
> > >
> > > Thank you very much for your positive recognition and valuable review and feedback, which help improve the quality of our work. We will include all discussions in the revision.

---

### Official Review · Reviewer_83c7 · 2026-03-13

**Soundness:** 3
**Presentation:** 3
**Significance:** 2
**Originality:** 2
**Overall Recommendation:** 3
**Confidence:** 3

**Summary:**

This paper introduces WinDeskGround, a benchmark designed to evaluate GUI grounding ability in complex multi-window desktop environments. The authors claim that realistic user working environments are multi-task parallel and complex in window arrangement, whereas current grounding benchmarks lack this complexity. They identify 9 categories from Windows applications, obtain screenshots and annotate them, thus constructing a meta-benchmark. Then they design three key parameters to dynamically construct the evaluation data: Layout Density, Semantic Similarity, and Occlusion Rate. Detailed experimental results on five leading grounding models show that occlusion is the most important factor affecting grounding ability in multi-window environments.

**Compliance With Llm Reviewing Policy:**

Affirmed.

**Final Justification:**

The authors address my main concerns. I decided to raise my score.

**Key Questions For Authors:**

1. I have doubts about the parameter choices for benchmark construction: the density level reaches 15? Each application is fixed to a specific area? Do the authors have support for these choices?
2. How did the authors ensure that the target element would not be hidden in the occlusion case? There is no explanation either in the text or in Algorithm 1.

**Limitations:**

yes

**Strengths And Weaknesses:**

Strengths:

- The paper is well written. All the baselines and experimental designs are clear, although with some small typos such as the way of using quotation marks.
- The results are significant. Especially for occlusion, the performance drop is linearly correlated with the occlusion ratio, nearly the same for all models, which supports the authors' claim that multi-window complex desktop environments are indeed difficult.
- The dynamic construction of the benchmark makes it suitable for different evaluation cases, making it useful for future development in the GUI agent community.

Weaknesses:

- Originality is not sufficient. The authors claim that the existing community lacks such multi-window desktop environments, yet ScreenSpot-Pro does have such data entries. This undermines the authors' claim and proves that the related work has not been thoroughly reviewed.
- According to Figure 4, occlusion takes the most responsibility for performance drop, whereas all models' performance on "Visual Clutter" and "Semantic Interference" remains nearly the same, which makes me doubt the validity of designing these two factors.
- The synthetic-to-real gap is not validated. The paper claims to simulate "realistic" desktops, but never validates whether performance on WinDeskGround correlates with performance on actual real desktop screenshots or computer-use benchmarks. Without this, it is unclear whether improvements on this benchmark transfer to real-world deployment.
- Instruction validity at 85% is concerning. For a benchmark paper, 15% of instructions being invalid is a non-trivial noise floor that could affect the reliability of results, especially for the harder difficulty levels where accuracy numbers are already very low (10–11% at L5).

---

> ### Author Rebuttal · Authors · 2026-03-30
>
> We sincerely thank the reviewer for the exceptionally detailed and highly constructive feedback. We value these concerns and provide point-by-point clarifications and supplements below.
> ### W1. Insufficient novelty (overlap with ScreenSpot-Pro)
>
> > A1. We sincerely thank the reviewer for this insightful and important observation. We fully agree that existing works (e.g., ScreenSpot-Pro) include multi-window samples, and we will clarify this. We would like to respectfully clarify that the key difference of our work is transforming the multi-window setting from a passive data distribution into an explicitly controllable variable.
> >
> > Specifically, we propose a parameterized synthesis framework that independently controls layout density, occlusion, and semantic similarity, enabling factor-level analysis. In contrast, ScreenSpot-Pro relies on manually collected data (≈60% multi-window, ≈20% non-foreground targets), limiting controlled analysis.Moreover, our structured composition improves scalability and controllability. Therefore, our contribution is not merely including multi-window samples, but establishing a controllable and decomposable evaluation paradigm.
>
> ### W2. Visual clutter and semantic interference seem ineffective
>
> > A2. We sincerely thank the reviewer for this insightful observation. We would like to clarify that the flatter curve mainly results from our controlled evaluation design: when analyzing one factor, the others are fixed to simple settings. They still induce stable drops compared to the single-window settingFor example, increasing window count to 3 or semantic similarity to level 3 causes consistent 5%–10% degradation across models (e.g., InfiGUI, OS-Atlas, UI-TARS). Thus, rather than ineffective, they are secondary yet stable factors for comprehensive evaluation. We will clarify this rationale and improve the visualizations.
>
> ### W3. The gap between synthetic data and real data is not verified
> > A3. We sincerely thank the reviewer for this important and insightful suggestion. We clarify that our data is not fully synthetic: UI elements and annotations come from real screenshots, while scenes are generated via controlled synthesis. Synthetic data is widely used for its controllability. Our goal is not to replace real data, but to provide a controllable testbed for analyzing factors such as occlusion and layout complexity. To assess realism, we conduct a perceptual evaluation using VLMs as judges. Results show a high realism score (86.68%), indicating the generated scenes are perceived as highly realistic. More details will be included in the revision.
>
> ### W4. The 85% instruction validity is concerning
> > A4. We sincerely thank the reviewer for raising this important concern. The 85% validity is measured under strict criteria; many rejected cases are boundary ambiguities rather than incorrect annotations. In addition, the sample contains more difficult cases (L3–L5), making this estimate conservative. We further expanded the validation across difficulty levels, with the results as follows:
> >
> > | Level   | bbox_accuracy | target_clickable | multiple_valid_targets |
> > | - | - | - | - |
> > | overall | 93.20%| 94.40%| 9.64%|
> > | L1| 99.00%| 99.20%| 4.60%|
> > | L2| 98.40%| 99.40%| 7.20%|
> > | L3| 97.40%| 98.00%| 9.40%|
> > | L4| 91.60%| 92.40%| 11.60%|
> > | L5| 79.60%| 83.00%| 15.40%|
> >
> > Expanded validation across difficulty levels shows high quality for L1–L3, while higher levels introduce ambiguity but remain useful as stress tests. We will clarify this definition and provide further explanation in the revision.
>
> ### Q1. Basis for parameter selection
> > A5. We sincerely thank the reviewer for this excellent question. The parameter ranges are designed to span from common usage to stress-testing scenarios, rather than to match a specific real-world distribution. Window density and layout priors are informed by HCI research (e.g., [1][2]), but mainly serve controlled analysis. We will clarify this design goal.
> >
> > [1] Towards ideal window layouts for multi-party, gaze-aware desktop video conferencing.
> >
> > [2] Psychologically-inspired, unsupervised inference of perceptual groups of GUI widgets from GUI images.
>
> ### Q2. How to ensure the target is not completely occluded
>
> > A6. We sincerely thank the reviewer for this excellent question. We enforce visibility constraints to prevent full occlusion. The target is projected to desktop coordinates, and a visible ratio$v \in [v_{min}, v_{max}]$is sampled to determine occlusion area. A distractor window is then positioned with controlled overlap and layering (background < target < occluder). After synthesis, we compute the actual visible ratio using a pixel-level mask and record it. Thus, targets remain partially visible rather than fully occluded. We will clarify this process in the revision.
>
> We hope our clarifications and additional evidence have fully addressed the reviewer’s concerns, and we would greatly appreciate a reconsideration of the score.

---

> > ### Author Rebuttal · Reviewer_83c7 · 2026-04-03
> >
> > Thank you for the detailed response. I have no further questions for now.

---

> > > ### Author Response · Authors · 2026-04-04
> > >
> > > Thank you very much for your reconsideration and increasing the score! We are glad that the rebuttal has fully addressed the concerns.

---

### Decision · Program_Chairs · 2026-04-30

**Decision:**

Accept (regular)

**Comment:**

WinDeskGround is a benchmark for testing computer-use agents in the presence of distracting windows. Most existing benchmarks focus only on single and clean windows. The practical appeal for this problem is high. They introduce a controllable environment where one can control three factors: occlusion, layout density and semantic similarity. The key finding is that occlusion results in a linear drop in accuracy as visibility decreases. It has 585 real Windows application screenshots across 9 application categories, which are then synthetically manipulated with the controlled factors. Reviewers broadly agreed the contribution is useful and the experiments with 5 models are thorough. The dataset could be valueable for improving CUA agents.